# Anti-tumor necrosis factor-alpha monoclonal antibody suppresses colorectal cancer growth in an orthotopic transplant mouse model

**Takeshi Takasago[1], Ryohei Hayashi [2]\*, Yoshitaka Ueno[2], Misa Ariyoshi[1], Kana Onishi[1], Ken Yamashita[2], Yuichi Hiyama[2], Hidehiko Takigawa[2], Ryo Yuge[1], Yuji Urabe[3], Shiro Oka[1], Yasuhiko Kitadai[4], Shinji Tanaka[2]**

1 Department of Gastroenterology, Hiroshima University Hospital, Hiroshima, Japan, 2 Department of Endoscopy, Hiroshima University Hospital, Hiroshima, Japan, 3 Gastroenterointesinal Endoscopy and Medicine, Hiroshima University Hospital, Hiroshima, Japan, 4 Department of Health and Science, Prefectural University of Hiroshima, Hiroshima, Japan

\* r-hayashi@hiroshima-u.ac.jp

**Data Availability Statement:** The data supporting the findings of this study are included in Supporting information and have been uploaded to the NCBI Gene Expression Omnibus. (https://www.

## Abstract

The risk of malignant tumor progression has been a concern associated with the use of anti-tumor necrosis factor-alpha monoclonal antibody (anti-TNFα mAb). On the contrary, recent observational studies have reported negatively on this risk and instead suggested that anti-TNFα mAb acts as a tumor suppressor in inflammatory carcinogenesis models and subcutaneous transplant models of colorectal cancer. However, no consensus has been established regarding the actual effects of anti-TNFα mAb on malignant tumors. Here, we aimed to evaluate, for the first time, the effect of anti-TNFα mAb on the tumor microenvironment in the absence of intestinal inflammation in a colorectal cancer orthotopic transplant mouse model suitable for tumor microenvironment assessment. The orthotopic transplantation model was developed by transplanting CT26 cells into the cecum of *BALB/c* mice. Changes in tumor size and weight were recorded 3 weeks after transplantation, and the tumor microenvironment was assessed via RNA sequencing and immunohistological staining. In the orthotopic transplant model, the administration of anti-TNFα mAb led to a reduction in colorectal cancer. The RNA sequencing analysis showed upregulation of immune-related pathways and apoptosis and suppression of stromal- and tumor growth-related pathways. Additionally, Gene Ontology analysis showed inhibition of angiogenesis. Immunohistochemical staining showed inhibition of tumor growth, increase in apoptosis, suppression of stromal response, suppression of angiogenesis, enhancement of tumor immunity, and reduction in the number of tumor-associated macrophages. Anti-TNFα mAb acts as an inhibitor of tumor progression in the tumor microenvironment of a colorectal cancer orthotopic transplant mouse model.

## Introduction

Anti-tumor necrosis factor-alpha monoclonal antibody (anti-TNFα mAb) suppresses inflammation by inhibiting the activity of TNFα and is employed as a major therapeutic agent for

ncbi.nlm.nih.gov/geo/query/acc.cgi?acc=
GSE226619) Accession number is GSE226619.

**Funding:** The author received no specific funding for this work.

**Competing interests:** The authors have declared that no competing interests exist.

immune-mediated inflammatory diseases such as inflammatory bowel disease (IBD) and rheumatoid arthritis (RA) [1]. Facilitating the formation of malignant tumors via immunosuppression is a matter of concern [2]. The European Crohn's and Colitis Organisation (ECCO) guidelines state that the use of anti-TNFα mAb can be detrimental to patients undergoing cancer treatment and should be avoided for 2 years (5 years for patients with a high risk of recurrence) after cancer treatment [3]. However, varied perspectives have been presented with respect to the actual effect of administering anti-TNFα mAb for the suppression of TNFα in patients with malignant cancers, and no consensus has been established.

Recent observational studies have reported negatively on the increased risk of promotion of malignancy in patients using anti-TNFα mAb [4–6]. In contrast, as sporadic reports of tumor growth during anti-TNFα mAb use have been published [7, 8], the possibility that anti-TNFα mAb use may influence tumor growth cannot be ruled out.

In recent years, anti-TNFα mAb has been increasingly used for the treatment of RA or IBD, such as ulcerative colitis and Crohn's disease. With an increase in the number of patients with IBD and RA in higher age groups [9, 10], it is assumed that more patients with carcinoma will be considered for the administration of anti-TNFα mAb. In addition, the administration of infliximab, an anti-TNFα mAb, for immune-related adverse events, such as colitis, associated with the administration of immune checkpoint inhibitors to patients with tumors is expected to increase in the future.

Barring a few studies, no prognostic effect of the administration of anti-TNFα mAb to patients with cancer has been observed in most studies [11]. It is difficult to clinically evaluate the effect of anti-TNFα mAb on tumors alone, as patients with IBD and RA are often administered other immunosuppressive agents. With respect to colorectal cancer (CRC), the effects of anti-TNFα mAb on tumors have been evaluated *in vitro*, in inflammatory carcinogenesis models, and in CRC subcutaneous allogeneic transplant models. The results of these studies have demonstrated possible anti-tumor effects of anti-TNFα mAb [12–14].

However, to the best of our knowledge, there are no reports on the effects of anti-TNFα mAb on orthotopic transplantation of CRC. An orthotopic transplant model reflects the tumor microenvironment (TME) and is recommended for assessing tumor progression, angiogenesis, invasion, stroma, and metastasis [15, 16], whereas subcutaneous xenografts do not reproduce the interactions between the tumor and its microenvironment [17]. Therefore, the orthotopic transplant model is a more suitable experimental system for evaluating TME than the subcutaneous xenograft model. Regarding the inflammatory carcinogenesis model, anti-TNFα mAb may suppress the growth of tumors by inhibiting intestinal inflammation [13, 14]. However, the actual effects of anti-TNFα mAb on tumors have not been fully evaluated owing to the substantial effect of inflammation.

In this study, we aimed to investigate the effects of anti-TNFα mAb on tumor progression and TME in the absence of intestinal inflammation using an allogeneic immune response orthotopic transplantation model of CT26, a CRC cell line derived from *BALB/c* mice.

## Materials and methods

### CRC cell lines and culture conditions

CT26 (American Type Culture Collection, Manassas, VA, USA), a clonal cell line derived from colon cancer cells of *BALB/c* mouse, was used as the representative of the CRC cell line.

We cultured CT26 cells in Dulbecco's modified Eagle medium (DMEM; Sigma-Aldrich; Merck KGaA, Germany) at 37˚C. DMEM was supplemented with 10% fetal bovine serum (FBS; Sigma-Aldrich, Merck KGaA) and a penicillin–streptomycin mixture. The cells were not cultured for more than 12 weeks after their recovery from the frozen stock.

## Evaluation of the effect of TNFα on cell proliferation *in vitro*

CT26 cells ($1.0 \times 10^4$ cells/well) were seeded and cultured in 24-well plates (ImageLock; Essen Bioscience) containing DMEM supplemented with 0.5% FBS. Recombinant mouse TNF-α (cat. no. 575202; Biolegend, San Diego, CA, USA) was added to DMEM to reach the target concentration. Bright-field images obtained using IncuCyte® Zoom (Essen Bioscience) and IncuCyte software (version 2015A Rev1; Essen Bioscience) automatically expressed cell confluence as a percentage over 2 days to generate a figure of cell proliferation. These experiments were performed in triplicate.

## Orthotopic transplantation of tumor cells into experimental animals

Female *BALB/c* mice were obtained from Charles River Japan (Tokyo, Japan) and raised under specific pathogen-free conditions until they reached 8 weeks of age. Animal experiments in this study were approved by the Animal Experiment Committee of Hiroshima University (A19-144). Anesthesia was induced via an intraperitoneal injection of medetomidine (0.3 mg/kg), midazolam (4 mg/kg), and butorphanol (5 mg/kg).

Transplantation was performed under a zoom stereomicroscope (Carl Zeiss, Oberkochen, Germany) by injecting $1 \times 10^4$ CT26 cells in 30 μL of Hank's balanced salt solution into the cecal wall of female *BALB/c* mice under anesthesia.

## Intraperitoneal administration of anti-TNFα mAb to mice

Anti-TNFα mAb (MP6-XT22, 1 mg/mouse; Biolegend, San Diego, USA) was administered to mice according to a previous study [14].

Mice subjected to orthotopic transplantation were intraperitoneally administered 1 mg of anti-TNFα mAb (MP6-XT22; BioLegend) every 7 days, as shown in Fig 1A (anti-TNFα mAb group, n = 8).

Phosphate-buffered saline (500 μL) was administered intraperitoneally to the mice in the control group with the same dosing schedule as that employed for mice in the anti-TNFα mAb group (control group, n = 12). All mice were then analyzed on day 21. The mice were sacrificed by cervical dislocation under deep sedation with the same anesthesia used for orthotopic transplantation.

## Necropsy procedures

At the time of analysis, the weight of the mice was recorded. After grossly confirming the presence of metastases in the liver, lungs, and lymph nodes, the tumors were excised, and their weight and length were measured. The volume of the tumor was calculated with the following formula: V = 1/2 (length × width$^2$). The resected tumors were fixed using a formalin-free immunohistochemistry zinc fixative (BD Pharmingen; BD Biosciences) for 24 h at room temperature. The tumors were then embedded in paraffin for immunohistochemical analysis.

## Immunohistological staining

In this experiment, the following primary antibodies were used: monoclonal rabbit anti-Ki67 (dilution, 1:300; cat. no. GTX16667; Genetex, Inc. Irvine, CA, USA), monoclonal rabbit anti-F4/80 (dilution, 1:500; cat. no. D2S9R; Cell Signaling Technology, Inc., Danvers, MA, USA), monoclonal rabbit anti-CD163 (dilution, 1:500; cat. no. ab182422; Abcam, plc., Cambridge, UK), polyclonal rabbit anti-CD31 (dilution, 1:300; cat. no. ab28364; Abcam, plc.), monoclonal rabbit anti-CD8α (dilution, 1:400; cat. no. D4W2Z; Cell Signaling Technology, Inc.), monoclonal rabbit anti-CD4 (dilution, 1:400; cat. no. D7D2Z; Cell Signaling Technology, Inc.),

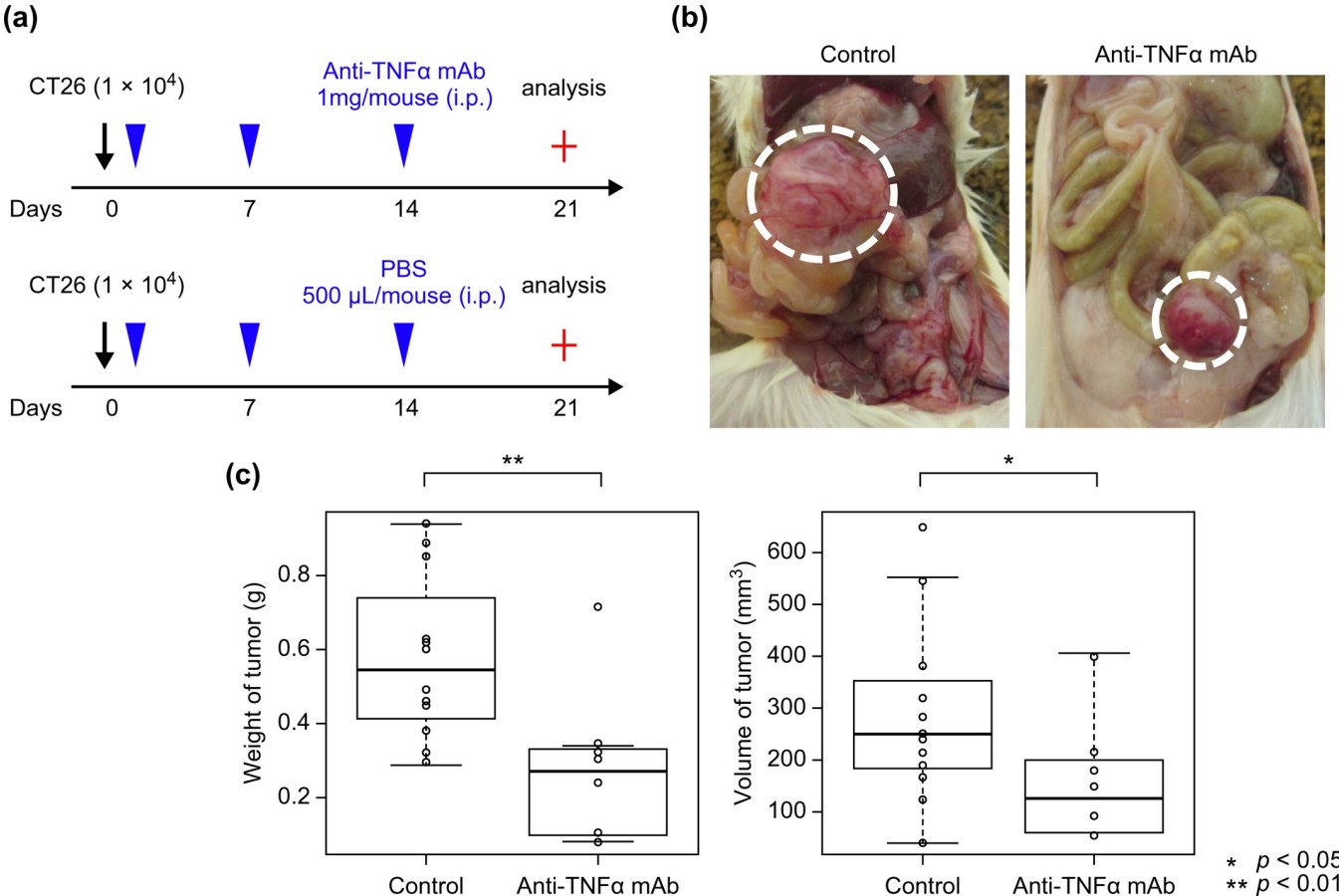

**Fig 1. Anti-TNFα mAb suppresses colorectal cancer growth in an orthotopic transplant model.** (a) Schematic overview of the orthotopic transplant model of colorectal cancer and anti-TNFα mAb administration. (b) Mice subjected to orthotopic transplantation were administered CT26 cells into the cecum with and without anti-TNFα mAb. (c) The weight and volume of the tumor orthotopically transplanted into the cecum were recorded. Control group (n = 12), anti-TNFα mAb group (n = 8). anti-TNFα mAb, anti-tumor necrosis factor-alpha monoclonal antibody.

polyclonal goat anti-mouse NKp46/NCR1 (dilution, 10 μg/mL; cat. no. AF2225; R&D Systems, Inc., Minneapolis, MN, USA), rabbit anti-alpha smooth muscle actin (α-SMA; dilution, 1:50; cat. no. ab5694; Abcam, plc.), polyclonal rabbit anti-mouse type I collagen (dilution, 1:500; cat. no. 20151; Novotec, Bron, France), and polyclonal goat anti-fibronectin (FN; dilution, 1:200; cat. no. sc-6952; Santa Cruz Biotechnology, Inc., Dallas, TX, USA). The following two secondary antibodies were used: polyclonal goat anti-rabbit immunoglobulin/biotinylated antibody (dilution, 1:500; cat. no. E0432; Dako, Glostrup Kommune, Denmark) and polyclonal rabbit anti-goat immunoglobulin/biotinylated antibody (dilution, 1:500; cat. no. E0466, Dako). Apoptosis was quantified using the terminal deoxynucleotidyl transferase dUTP nick-end labeling (TUNEL) assay with the ApopTag Plus Apoptosis Detection kit (#S7101; Millipore-Chemicon International, Temecula, CA).

The fixed tumor tissue was cut into sections of 4 μm thickness and used for immunostaining. Immunostaining was performed to examine CD8, CD4, NKp46/NCR1, αSMA, type 1 collagen, fibronectin, CD31, F4/80, and CD163 according to a previously reported protocol [18]. Ki67 labeling and apoptosis indices were evaluated with reference to a previous report [19]; these indices were defined as the ratio of positive cells to the total number of cells counted; they are expressed as percentage. All images were captured using an all-in-one fluorescence microscope (BZ-X710, KEYENCE, Osaka, Japan).

Quantification was randomly performed in five different microscopic fields of view (magnification, 400×). Positive cell counts, positive areas, and the total number of cells were automatically evaluated using a hybrid cell count application (BZ-H3C; KEYENCE, Osaka, Japan) with BZ-X Analyzer software (BZ-H3A; KEYENCE).

## RNA sequencing (RNA-Seq) analysis

Tumors (control group and anti-TNFα mAb group) that developed post-orthotopic transplantation were mechanically isolated using a homogenizer. RNA extraction was then performed using the RNeasy Mini kit (Qiagen, Hilden, Germany) according to the manufacturer's instructions. Library construction and data processing were performed by Qiagen (Hilden, Germany).

The following library preparation kits and sequencing equipment manufacturer's protocols were used: for poly(A) RNA preparation, Poly(A) mRNA Magnetic Isolation Module (New England Biolabs, Ipswich, MA, USA); for library preparation, NEBNext® Ultra™II Directional RNA Library Prep Kit for Illumina® (New England Biolabs, Ipswich, MA, USA); for sequencing, NovaSeq 6000 (Illumina Inc., San Diego, CA).

The quality of the raw paired-end sequence reads was assessed with FastQC (Version 0.11.7; https://www.bioinformatics.babraham.ac.uk/projects/fastqc/). Low-quality (< 20) bases and adapter sequences were trimmed using Trimmomatic software (Version 0.38) with the following parameters: ILLUMINACLIP: path/to/adapter.fa:2:30:10 LEADING:20 TRAILING:20 SLIDINGWINDOW:4:15 MINLEN:36. The trimmed reads were aligned to the reference genome using RNA-seq aligner HISAT2 (Version 2.1.0). The HISAT2-resultant.sam files were converted to.bam files using samtools and were used to estimate the abundance of uniquely mapped reads with featureCounts (Version 1.6.3). The raw counts were normalized with transcripts per million (TPM). Based on the normalized read counts, we conducted comparative analyses of samples.

Differentially expressed genes (DEGs) were detected using DESeq2 (Version 1.24.0) with the threshold of |log2FC (Fold Change)| > 1 and p.adjust < 0.05 calculated using the Benjamini and Hochberg (BH) method. Gene set enrichment analysis was conducted using DAVID (Version 1.22.0) and gene ontology (GO) terms with p.adjust < 0.05 by the BH method were extracted. (Fig 2A and 2B) DEG data are presented in S1 Appendix. DEGs were analyzed by Ingenuity Pathway Analysis (IPA) (Version 51963813; QIAGEN Inc., https://www.qiagenbioinformatics.com/products/ingenuity-pathway-analysi) [20]. Canonical pathways are expected to be "increased or decreased" on analyzing data on changes in gene expression, relative to known databases. Pathways with positive z-scores (indicated in orange) were predicted to increase activity, whereas those with negative z-scores (indicated in blue) were predicted to decrease activity. Pathways were classified into three categories according to their functions: immune response, stromal reaction, and cell proliferation (Fig 2C).

In the network of regulator effects, the functions and diseases predicted to be affected by gene-expression changes indicated by green circles, when visualized as a network (Fig 2D).

These data were obtained by Rhelixa (Tokyo, Japan).

## Statistical analysis

Significant differences in tumor weight and volume and the immunostaining results of the control and anti-TNFα mAb groups were assessed using Mann–Whitney U test. Differences were considered statistically significant at $P < 0.05$.

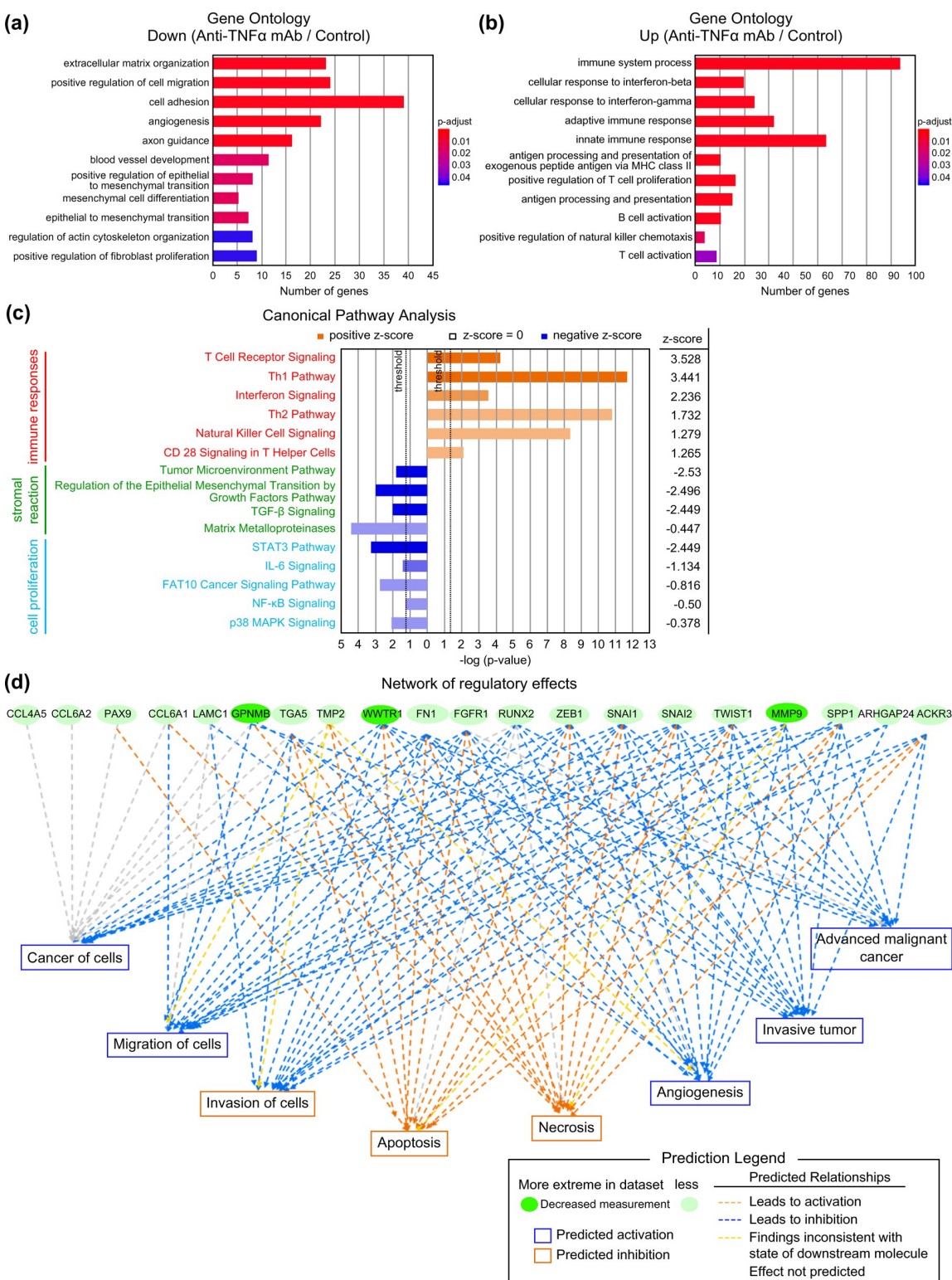

**Fig 2. Summary of RNA sequencing analysis of the anti-TNFα mAb group and comparison with the control group.** (a) GO terms that were downregulated in the anti-TNFα mAb group compared with those in the control group. Significant changes were defined with adjusted *P*-values of less than 0.05. Control group (n = 3), anti-TNFα mAb group (n = 3). (b) GO terms that were enriched in the anti-TNFα mAb group compared with those in the control group.Significant changes were defined with adjusted *P*-values of less than 0.05. Control group (n = 3), anti-TNFα mAb group (n = 3). (c) Canonical pathways identified using IPA that were upregulated or

downregulated in the anti-TNFα mAb group compared with those in the control group. Extracted pathways are shown on the horizontal axis (threshold set at *P*-value < 0.05). Control group (n = 3), anti-TNFα mAb group (n = 3). (d) Network of regulator effects in the anti-TNFα mAb group compared with those in the control group observed using IPA. Control group (n = 3), anti-TNFα mAb group (n = 3). anti-TNFα mAb, anti-tumor necrosis factor-alpha monoclonal antibody; GO, Gene Ontology; IPA, ingenuity pathway analysis.

## Results

### Evaluation of the effect of TNFα on cell proliferation

The proliferation ability of CT cells when cultured with TNFα (100 pg/mL). Compared to that in the controls, significant cell proliferation was observed at 48 h (S1A Fig). Similar results were observed when cultured with TNFα at 1, 10, and 1000 pg/mL (data not shown). Regarding the proliferation ability of CT cells when cultured with TNFα (25, 50, 100, or 200 ng/mL) compared with that of the control, there was no significant difference with TNFα administration (S1B Fig).

### Administration of anti-TNFα mAb reduced the size of CT26 tumor in mice

Mice were subjected to orthotopic transplantation, and 1 mg of anti-TNFα mAb was administered intraperitoneally every 7 days as shown in Fig 1A. In the anti-TNFα mAb group, a macroscopic reduction in tumor size was observed compared with that in the control group, along with a significant reduction in tumor weight and volume (Fig 1B and 1C). No apparent lymph node or liver metastases were observed in both groups. No significant difference was observed in the body weight of mice during at analysis (Table 1, S2 Fig).

### Immune response was enhanced, and stromal reaction, angiogenesis, and cell proliferation were suppressed

To investigate the effect of anti-TNFα mAb on gene expression in transplanted tumors, mRNA levels were analyzed, and GO and IPA analyses were used to comprehensively examine the differences in gene expression and pathways. The GO analysis showed that GO terms such as "extracellular matrix organization," "positive regulation of cell migration," "angiogenesis," and "epithelial to mesenchymal transition" were significantly downregulated in the group treated with anti-TNFα mAb compared with those in the control group (Fig 2A). The GO terms related to immune response, such as "T cells" and "interferons," were significantly enriched in the group treated with anti-TNFα mAb compared with those in the control group (Fig 2B).

The canonical pathways extracted using the IPA predicted the association of the group treated with anti-TNFα mAb with enriched T cell-related pathways and pathways involved in

**Table 1. Results of experiments performed with mice orthotopically transplanted with colorectal cancer cells and treated with anti-TNFα mAb.**

| Group | No. | Body weight, before g (range) | Body weight, after g (range) | Tumor weight, g (range) | Tumor volume, mm³ (range) | Lymph node metastasis, n | Liver metastasis, n |
|---|---|---|---|---|---|---|---|
| Control | 12 | 22.0 (19.0–24.0) | 20.0 (16.0–25.0) | 0.55 (0.29–0.94)** | 250.5 (125–650)* | 0/12 | 0/12 |
| Anti-TNFα mAb | 8 | 21.5 (19.0–23.0) | 20.5 (15.0–23.0) | 0.27 (0.08–0.71) | 125.0 (62.5–405) | 0/8 | 0/8 |

Median (range), anti-tumor necrosis factor-alpha monoclonal antibody (anti-TNFα mAb)

*$P < 0.05$ vs. anti-TNFα mAb group

**$P < 0.01$ vs. anti-TNFα mAb group

immune response, such as "interferon signaling" and "natural killer cell signaling" compared with those in the control group. In the canonical pathway analysis, pathways related to stromal reaction, such as "tumor microenvironment pathway," "regulation of the epithelial mesenchymal transition by growth factor pathway," "TGF-β signaling," and "inhibition of matrix metalloproteinase" were expected to be suppressed, and pathways related to tumor growth, such as "inhibition of signal transducer and activator of transcription (STAT3) pathway," "IL-6 signaling," "FAT10 cancer signaling pathway," "NF-κB signaling," and "p38 mitogen-activated protein kinase (MAPK) signaling" were expected to be inhibited (Fig 2C). The main figure of the network of regulator effects showed "migration of cells," "invasive tumor," "proliferation of cells," "advanced malignant cancer," and "apoptosis" through the suppression of genes such as *MMP9* and *FN1* (Fig 2D).

These results suggest that the administration of anti-TNFα mAb suppresses tumor-associated pathways. Anti-TNFα mAb may inhibit tumor progression by suppressing cell proliferation, enhancing immunity against tumors, increasing apoptosis, and suppressing stromal reaction and angiogenesis. Based on these results, additional evaluation was performed with immunostaining.

### Anti-TNFα mAb suppressed tumor growth and increased apoptosis

To assess the effect of anti-TNFα mAb, orthotopically implanted tumors were analyzed immunohistochemically. The Ki67-labeling index was significantly decreased in the anti-TNFα mAb group, indicating that tumor growth was suppressed by anti-TNFα mAb (Fig 3A and 3B).

Apoptosis was evaluated using the TUNEL method. The results showed that apoptosis was enhanced in the anti-TNFα mAb group. This finding is consistent with the upregulation of apoptosis observed in the RNA-Seq analysis (Fig 3A and 3B).

### Anti-TNFα mAb suppressed stromal reaction and angiogenesis

Stromal reaction was evaluated using immunostaining for αSMA, type 1 collagen, and fibronectin. The expression of αSMA, type 1 collagen, and fibronectin was significantly downregulated in the anti-TNFα mAb group, indicating that the stromal reaction was suppressed in this group. These results were consistent with the suppression of stromal reaction observed in the RNA-Seq analysis (Fig 4A and 4B). Immunostaining of CD31 was also performed to evaluate angiogenesis, and the vascular region was found to be significantly reduced in the anti-TNFα mAb group, indicating that angiogenesis was suppressed in this group. This result was consistent with the inhibition of angiogenesis observed in the RNA-Seq analysis (Fig 4A and 4B).

### Anti-TNFα mAb enhanced tumor immunity and suppressed tumor-associated macrophages

Immunostaining of CD8, CD4, and NKp46/NCR1 (used to evaluate natural killer cells) was performed to assess immunity against the tumor. The number of cells that were positive for the expression of CD8 and NKp46/NCR1 was significantly increased in the anti-TNFα mAb group, indicating that immunity against the tumor was enhanced in the anti-TNFα mAb group. The expression of CD4 did not differ between the groups, which was inconsistent with the RNA-Seq results (Fig 5A and 5B).

F4/80 (a pan-macrophage marker) and CD163 (an M2 macrophage marker) were used in immunostaining to analyze tumor-associated macrophages [21, 22]. The expression of both F4/80 and CD163 was significantly reduced in the anti-TNFα mAb group, indicating a decrease in the number of TAM (Fig 5A and 5B).

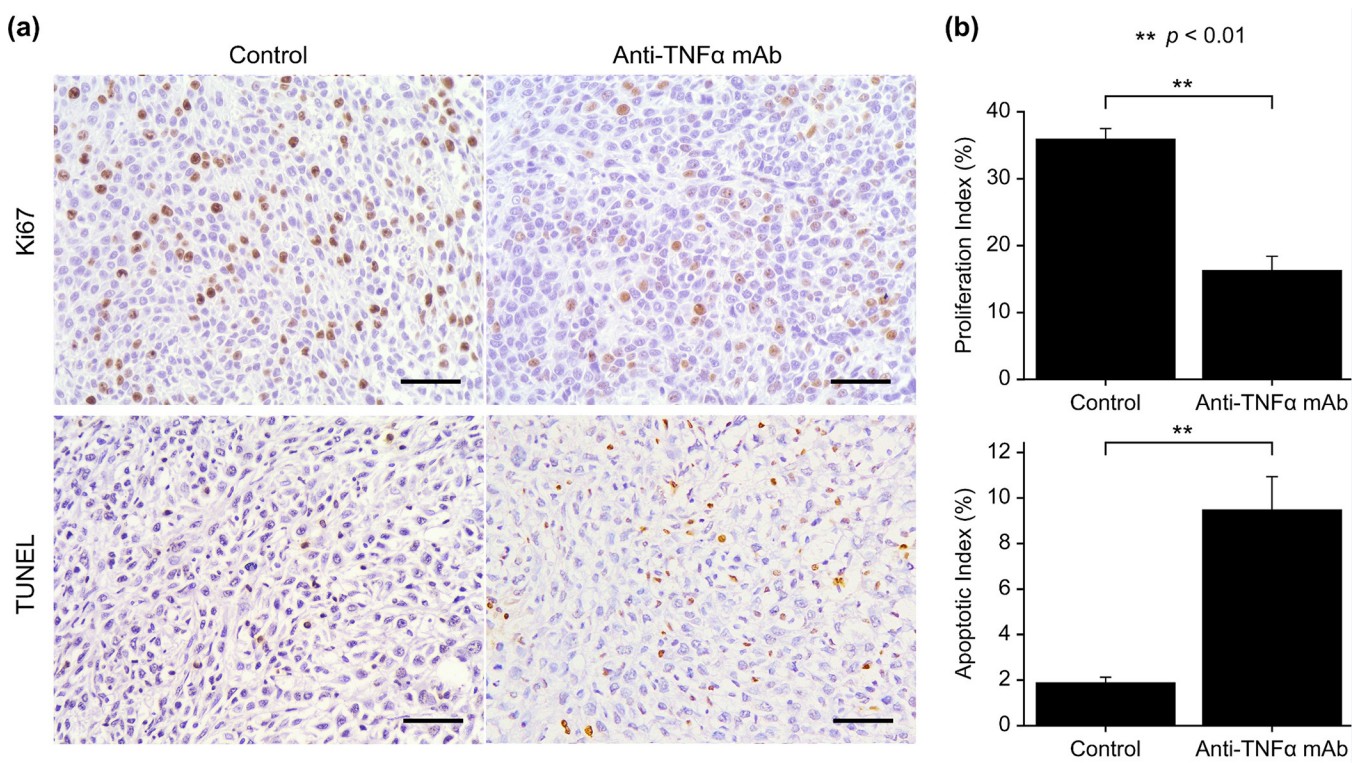

**Fig 3. Immunohistochemical analysis for examining cell proliferation and apoptosis in transplanted tumors.** (a and b) The number of cells stained positive for Ki67 was reduced in the anti-TNFα mAb group. The number of cells positive for TUNEL increased in the anti-TNFα mAb group. Data are presented as mean ± SEM. **$P < 0.01$. Scale bar, 50 μm. Control group (n = 12), anti-TNFα mAb group (n = 8). TUNEL, terminal deoxynucleotidyl transferase dUTP nick end labeling; anti-TNFα mAb, anti-tumor necrosis factor-alpha monoclonal antibody.

In summary, the results demonstrated immunohistological changes, such as the suppression of cell proliferation, enhancement of immunity against tumors, enhancement of apoptosis rate, suppression of stromal reaction, and suppression of angiogenesis, in the anti-TNFα mAb group.

## Discussion

Studies on CRC have investigated the effects of anti-TNFα mAb on tumors *in vitro* and in CRC subcutaneously implanted models [12–14]. Although the orthotopic model is an excellent model for evaluating the TME [15–17], to the best of our knowledge, no study has examined the effect of anti-TNFα mAb on tumors using this model. This study is the first to examine the effect of anti-TNFα mAb on the TME using such a CRC orthotopic transplantation model. This enabled us to more comprehensively investigate the TME, including tumor growth, stromal reaction, angiogenesis, TAM, immunity against tumors, and apoptosis, compared to previous studies.

TNFα is a cytokine that was initially identified to be responsible for tumor necrosis [23]. Previous studies have suggested that high doses of TNFα administered directly into the tumor induce tumor necrosis. However, in recent years, small doses of TNFα, such as those produced by the tumor, have been considered a possible tumor-promoting agent [24].

Low doses of TNFα promote tumor growth and angiogenesis in mouse models of melanoma, lung cancer, and mammary tumors [25]. In CRC, the actual TNFα concentration in colon cancer tissue is reportedly as low as 150 pg/mL [12]. Low concentrations of TNFα are

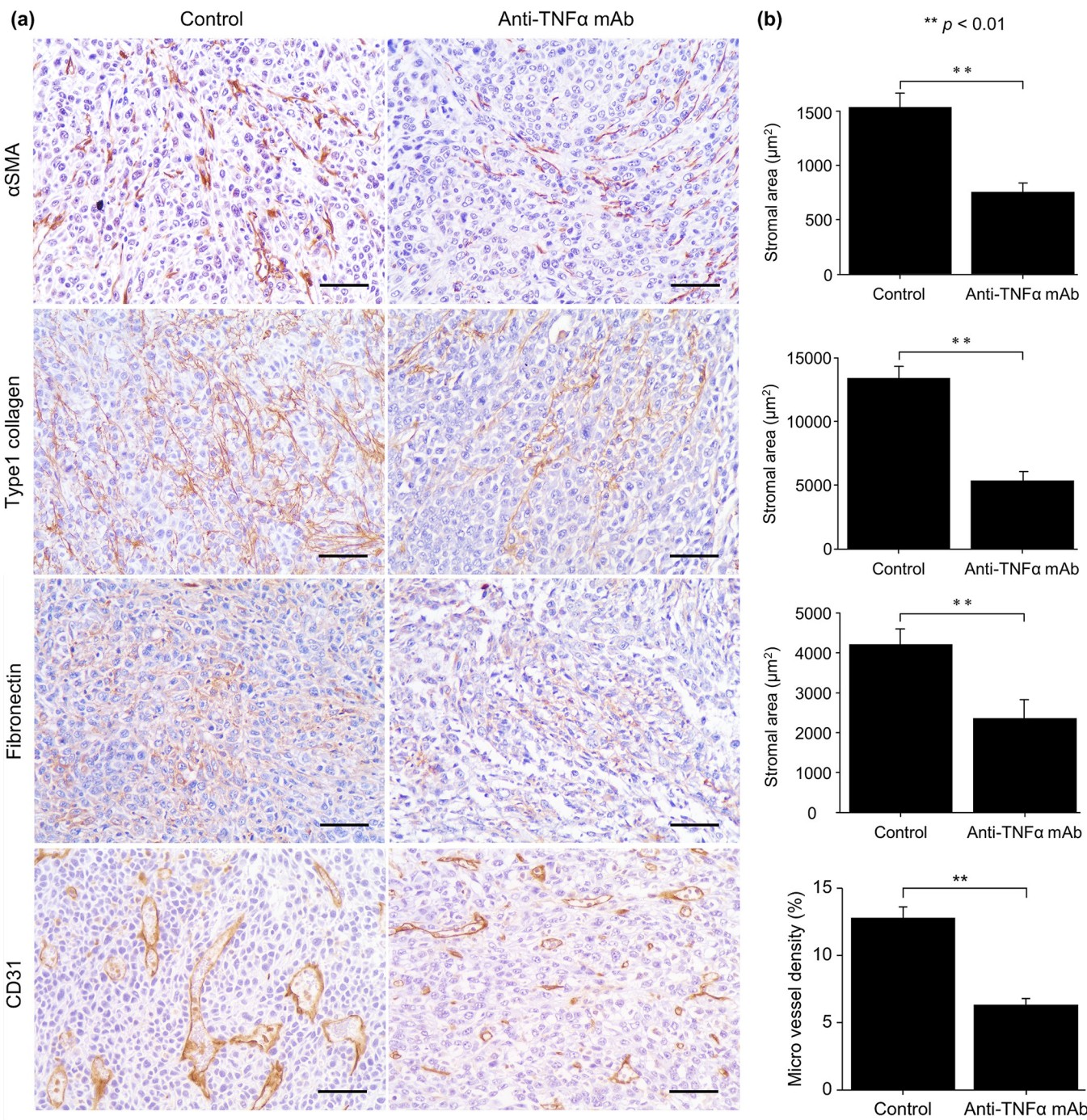

**Fig 4. Immunohistochemical analysis of transplanted tumors for examination of stromal reaction and angiogenesis.** (a and b) The area positive for αSMA, type1 collagen, fibronectin, and CD31 decreased in the anti-TNFα mAb group. Data are presented as mean ± SEM. $^*P < 0.05$. $^{**}P < 0.01$. Scale bar, 50 μm. Control group: n = 12 (×5 fields), anti-TNFα mAb group: n = 8 (×5 fields). αSMA, α-smooth muscle actin; anti-TNFα mAb, anti-tumor necrosis factor-alpha monoclonal antibody.

known to enhance the migration and invasion of CRC cells, such as CT26 [26]. In our study, low concentrations of TNFα increased tumor proliferation, similar to the results of the above-mentioned studies. In contrast, higher concentrations of TNFα did not result in an increase in tumor growth or a clear decrease in cell proliferation (S1 Fig). These results suggest that TNFα

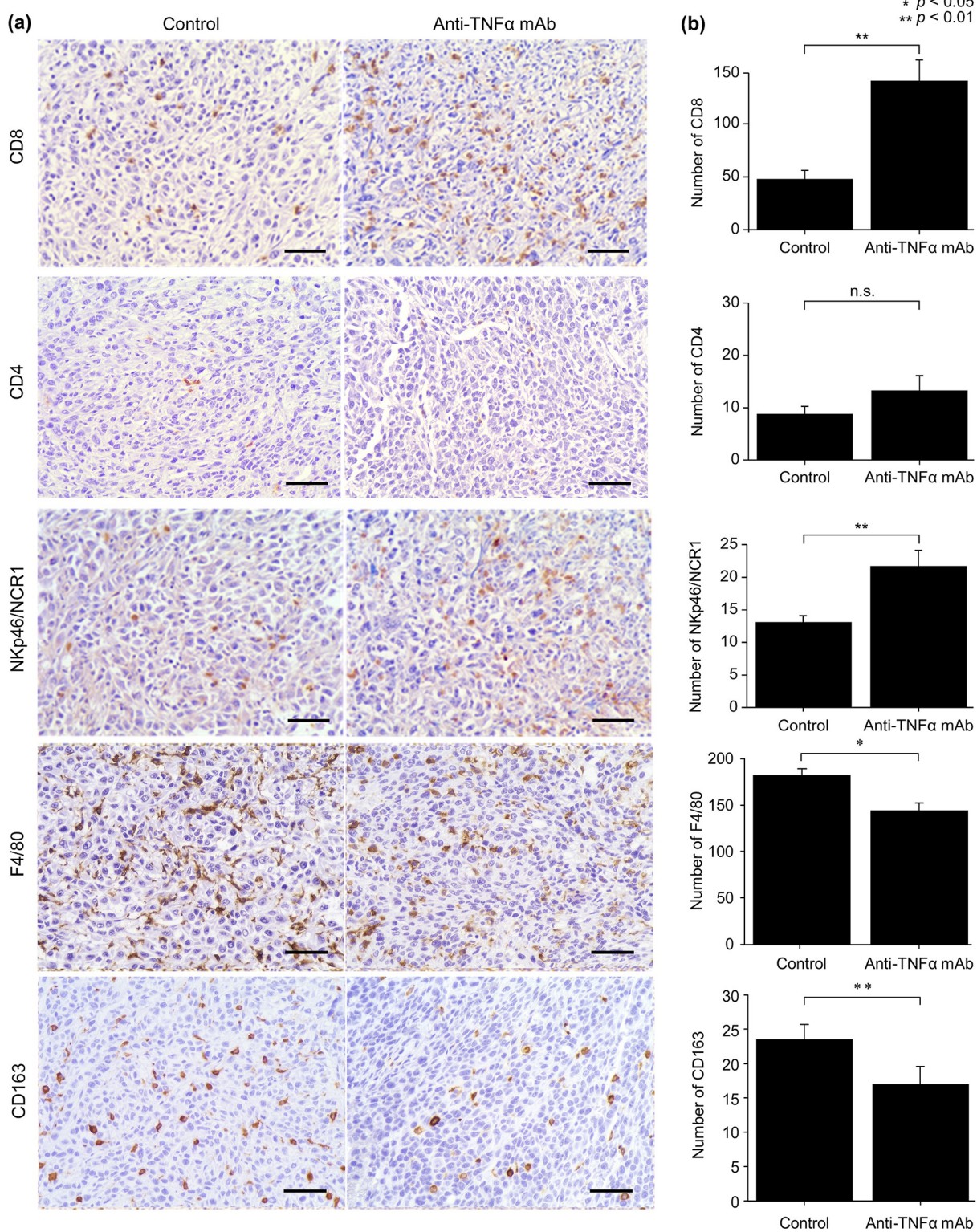

**Fig 5. Immunohistochemical analysis of transplanted tumors for immune response and tumor-associated macrophages.** (a and b) The number of cells positive for CD8 and NKp46/NCR1 was reduced in the anti-TNFα mAb group. The number of cells positive for CD4 did not differ between the groups. The expression of F4/80 and CD163 was reduced in the anti-TNFα mAb group. Data are presented as mean ± SEM. * $P < 0.05$. **$P < 0.01$. Scale bar, 50 μm. Control group (n = 12), anti-TNFα mAb group (n = 8). anti-TNFα mAb group, anti-tumor necrosis factor-alpha monoclonal antibody.

at endogenous tumor levels may aid in the promotion of tumor growth. Indeed, TNFα is a key mediator of cancer-related inflammation and is involved in tumor progression in the TME [24, 27–29]. Inhibition of TNFα also aids in the suppression of various carcinomas [30–33].

In CRC, high expression of TNFα is a poor prognostic factor. TNFα is highly expressed in *CT26* strains and human CRC, and the anti-tumor effect is enhanced upon administering anti-TNFα mAb in combination with oxaliplatin and 5-fluorouracil [12, 34, 35]. The administration of TNFα for the treatment of CT26 tumors affects tumor cell growth *in vitro* and tumor growth *in vivo* (subcutaneous transplantation) [36]. Therefore, suppressing the activity of TNFα may exert an anti-tumor effect.

In the present study, using an orthotopic transplantation model, we demonstrated that the administration of anti-TNFα mAb inhibits tumor growth, and the underlying mechanisms involves the suppression of stromal response, enhancement of tumor immunity, inhibition of angiogenesis, and increased apoptosis.

Under *in vitro* and *in vivo* (subcutaneous transplantation) conditions, TNFα mediates an increase in tumor growth, and inhibition of the activity of TNFα can lead to a reduction in tumor growth [36, 37]. In line with the previous study results, tumor growth was suppressed by anti-TNFα mAb in the orthotopic transplantation model used in this study.

Regarding the mechanism by which TNFα influences tumor growth, it is known that in CRC, one of the receptors of TNFα, namely TNFR2, regulates Ki67 expression, affecting fibroblast-associated proteins and αSMA, thereby increasing cell growth and migration [38].

In this study, apoptosis was enhanced in the anti-TNFα mAb group. Here, anti-TNFα mAb may have acted as a tumor suppressor. The suppression of NF-κB by anti-TNFα mAb inhibited the activity of antiapoptotic factors and promoted tumor apoptosis in a mouse liver model [27]. In a subcutaneous transplant model of melanoma and lung cancer, knocking out the gene encoding TNF receptor 2, the site of action of TNFα, enhanced apoptosis and suppressed tumor growth [39].

A study performed using colon cancer organoids has shown that TNF-α acts as a major factor mediating endothelial–mesenchymal transition (EMT) in the TME [40, 41]. TNF-α and TGF-β induce EMT via NF-κB in CRC cells, promoting CRC invasion and metastasis [42]. TNF-α is also known to promote TGF-β-induced EMT in human endothelial cells and oral squamous carcinoma-derived cells by enhancing TGF-β signaling. It also promotes cancer-associated fibroblast formation [43].

Matrix metalloproteinases (MMPs) have been implicated in EMT and angiogenesis in tumors [44]. MMP-9, which was found to be suppressed in the RNA-Seq analysis in this study, is known to promote tumor invasion and angiogenesis by activating TGF-β in breast cancer cells [45]. Its expression is induced by tumor-derived TNFα in breast cancer cells [46]. We speculate that the administration of anti-TNFα mAb may have suppressed the expression of MMP-9 and TGF-β, which may have inhibited angiogenesis and EMT.

TNFα induces the downregulation of NCR1/NKp46 in NK cells [47]. In esophageal cancer, TNFα suppresses NK cell function via NF-κB [48]. In melanoma, TNFα inhibition promotes CD8 infiltration into the tumor [49]. As TNF signaling suppresses the functions of CD8 and NK cells, which mediate anti-tumor effects, TNF inhibition may promote their functions, thereby leading to anti-tumor effects. In this study, we observed a few cells positive for CD4 expression; however, no significant difference was observed between the groups. TAMs, which are involved in tumor progression and angiogenesis, were also evaluated immunohistochemically, as TAMs are known to exhibit the M2 phenotype. TNFα acts on tumor growth and macrophage proliferation in CRC cells, and subcutaneous transplantation of CRC cells suppresses the expression of TNFα, resulting in a reduction in the size of tumor and the number of macrophages [37].

The MAPK pathway is one of the pathways involved in differentiation into M2 macrophages, and inhibition of this pathway suppresses the differentiation into M2 macrophages. This mechanism may also inhibit tumor suppression and angiogenesis [50].

This study had some limitations. First, although the short-term effects of anti-TNFα mAb on tumors can be evaluated in basic research, including this study, the long-term effects of anti-TNFα mAb on tumors have not been evaluated. Second, we used only one colon cancer cell line, CT26, and the effects of anti-TNFα mAb on other carcinomas and other colon cancer cell lines were not studied. Therefore, it cannot be concluded that anti-TNFα mAb is effective against other organ cancers and CRC based only on the results of this study. Although CT26 is known to have a high mutation rate and high immunogenicity, it is interesting to compare treatment outcomes in less immunogenic tumors such as MC-38 and spontaneous models of colorectal cancer; this is a subject for future study.

Third, this study showed significant changes in immune responses in the tumor microenvironment; however, these findings have not been evaluated in detail, such as by flow cytometry. we believe that these are vital future research topics.

In conclusion, anti-TNFα mAb demonstrated the potential to suppress tumor growth, stromal reaction, and angiogenesis, mediate a reduction in the number of TAMs, increase immunity against tumor and apoptosis, and suppress CRC progression in the TME. However, further investigations in other cell lines and cancer cell models are required to validate this conclusion. Moreover, it remains unclear whether anti-TNFα mAb is clinically safe in humans. Therefore, additional research on the effects of the use of anti-TNFα mAb on tumors should be conducted to clarify these issues.

## Supporting information

**S1 Fig. Evaluation of the effect of TNFα on cell proliferation.** (a) Proliferation ability of CT cells when cultured with TNFα (100 pg/mL). Compared to that in controls, significant cell proliferation was observed at 48 h. Similar results were observed on culture with TNFα at 1, 10, and 1000 pg/mL (data not shown). $^*P < 0.05$. (b) Proliferation ability of CT cells on culture with TNFα (25–200 ng) compared with that of the controls; there was no significant difference with TNFα administration (25, 50, 100, or 200 ng/mL). TNFα, tumor necrosis factor-alpha.
(TIF)

**S2 Fig. Percentage body weight change of mice in control group and anti-TNFαmAb group.** This figure shows the rate of weight change compared to DAY 0.
(TIF)

**S1 Appendix. Data of differentially expressed genes (DEGs).**
(XLSX)

## Acknowledgments

We would like to thank Editage (www.editage.jp) for English language editing.

A part of this study was performed at the Natural Science Center for Basic Research and Development, Hiroshima University.

## Author Contributions

**Conceptualization:** Takeshi Takasago, Ryohei Hayashi, Yoshitaka Ueno, Yasuhiko Kitadai.

**Data curation:** Takeshi Takasago, Misa Ariyoshi, Kana Onishi, Hidehiko Takigawa.

**Formal analysis:** Takeshi Takasago, Ryohei Hayashi, Yoshitaka Ueno, Ken Yamashita, Ryo Yuge, Yuji Urabe.

**Funding acquisition:** Ryohei Hayashi, Shinji Tanaka.

**Investigation:** Takeshi Takasago, Ryohei Hayashi, Misa Ariyoshi, Kana Onishi.

**Methodology:** Ryohei Hayashi, Yoshitaka Ueno, Yuichi Hiyama, Hidehiko Takigawa, Ryo Yuge, Yuji Urabe.

**Supervision:** Shiro Oka, Yasuhiko Kitadai, Shinji Tanaka.

**Visualization:** Takeshi Takasago, Ryohei Hayashi, Yoshitaka Ueno, Ken Yamashita, Yuichi Hiyama, Hidehiko Takigawa, Ryo Yuge, Yuji Urabe.

**Writing – original draft:** Takeshi Takasago, Ryohei Hayashi.

**Writing – review & editing:** Takeshi Takasago, Ryohei Hayashi, Yoshitaka Ueno.

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
