## [Decision Letter · Decision Letter 0]

9 Feb 2023

PONE-D-23-01681Anti-tumor necrosis factor-alpha monoclonal antibody suppresses colorectal cancer growth in an orthotopic transplant mouse modelPLOS ONE

Dear Dr. Hayashi,

Thank you for submitting your manuscript to PLOS ONE. After careful consideration, we feel that it has merit but does not fully meet PLOS ONE’s publication criteria as it currently stands. Therefore, we invite you to submit a revised version of the manuscript that addresses the points raised during the review process.

We look forward to receiving your revised manuscript.

Kind regards,

Kenji Fujiwara, Ph.D., M.D.

Academic Editor

PLOS ONE

2. To comply with PLOS ONE submissions requirements, in your Methods section, please provide additional information regarding the experiments involving animals and ensure you have included details on (1) methods of sacrifice and (2) efforts to alleviate suffering.

Additional Editor Comments:

Dear Dr. Hayashi.

Thank you for submitting your valuable manuscript to PLOS ONE. I think this manuscript is eligible to proceed toward the status of major revision. Please refer to the reviewers' comments. We will look forward to resubmission.

Best regards,

Kenji Fujiwara

Academic editor

Reviewers' comments:

Reviewer's Responses to Questions

**Comments to the Author**

1. Is the manuscript technically sound, and do the data support the conclusions?

Reviewer #1: Partly

Reviewer #2: Yes

2. Has the statistical analysis been performed appropriately and rigorously? 

Reviewer #1: Yes

Reviewer #2: Yes

3. Have the authors made all data underlying the findings in their manuscript fully available?

Reviewer #1: Yes

Reviewer #2: Yes

4. Is the manuscript presented in an intelligible fashion and written in standard English?

Reviewer #1: No

Reviewer #2: Yes

5. Review Comments to the Author

Reviewer #1: This manuscript provides an interesting investigation into the effects of anti-TNF-a treatment of colorectal cancer in an orthotopic mouse model. However, several concerns with the manuscript need to be addressed.

1. Only a single high dose of anti TNF antibody was used (1mg/mouse) both in this manuscript and in the publication cited as a reference for this method. Was a dose response experiment performed in vivo to determine the effective dose of MP6-XT22?

2. The method and results sections for RNA sequencing and analysis lacks important details. What methods were used to process sequencing reads, align them to a reference genome, and count the reads aligning to genes? How many genes were differentially expressed in the treatment vs control groups? What statistical method was used to determine whether changes in gene expression were significant?

3. Why was Mann–Whitney U test chosen for all statistical analysis? Were the assumptions of parametric tests violated by skew or unequal variance in the data?

4. Body weight is provided at endpoint showing no difference between control and anti-TNF groups but weight at day 0 is not provided. Plotting body weight at multiple time points over the course of the experiment would allow for visualization of any weight loss occurring during tumor growth.

5. Manuscript needs to be edited for spelling and grammar.

6. The legend for figure 2 is overly detailed. Some information provided in the fig 2 legend would be more appropriate in the results or methods section.

7. Sequencing raw data and/or comprehensive DEG dataset should be uploaded to a repository such as the NCBI Gene Expression Omnibus Database to facilitate access.

8. The results section describing figure 2 C and D is difficult to understand. Were these pathways “expected to be inhibited or suppressed” or were they found to be inhibited based on analysis of the data? Which specific gene’s expression changes led to the conclusion that each of these pathways are inhibited?

9. Figure 5, the images shown in A do not appear to be accurately represented by the graphs shown in B. This is especially evident in the F4/80 staining where control appears much greater than TNF but the graph only shows a 15% change. Why are the different stainings imaged at different magnifications as evidenced by the change in size of the scale bar?

10. The methods state that “quantification was randomly performed in 5 different fields of view” but the legend for figure 4 states that control had an n of 12 and TNF had an n of 8. It should be clarified whether each n represents an individual mouse, tumor section, or field of view.

11. The conclusion rightfully identifies several limitations with the study however these limitations should be discussed in more detail. CT26 is known to have a high mutation rate and be highly immunogenic. It would be interesting to compare the results of the treatment in a more poorly immunogenic tumor such as MC-38 or in a spontaneous model of colon cancer. Furthermore, the results of the present study appear to indicate significant changes in the immune response within the tumor microenvironment however, additional experiments to better characterize these changes would greatly strengthen the impact of the paper such as flow cytometry, immune-phenotyping of T cells and macrophages infiltrating the tumor, or cytokine expression array to quantify cytokine levels within the tumor tissue. As it stands, only very limited conclusions can be drawn from the data presented.

Reviewer #2: The authors have evaluated anti-tumor efficacy of anti-TNFα mAb against CT26 orthotopic model.

The results indicated tumor reduction in anti-TNFα mAb received mice. RNA sequencing analysis showed up-regulation of immune-related pathways and apoptosis and suppression of stromal- and tumor growth-related pathways in the tumor microenvironment (TME).

In addition, immunohistochemistry of tumors showed up-regulated infiltration of CD8 cells in TME. TUNEL assay indicated up-regulation of apoptosis in tumors.

Tumor-associated macrophages (TAMs) were analyzed using F4/80 and CD163 as markers. The results indicated TAMs suppression in TME.

Comments:

1) Legends for figures are scattered in the manuscript, please arrange it.

2) Legend for Fig 1 is missing! Please include it.

3) The results of evaluation of the effect of TNFα on cell proliferation is not mentioned in results section.

4) The authors showed down-regulation of TAMs in TME. What about M1 macrophages? Did they show up-regulation?

6. PLOS authors have the option to publish the peer review history of their article (what does this mean?). If published, this will include your full peer review and any attached files.

Reviewer #1: No

Reviewer #2: No

---

## [Author Response · Author response to Decision Letter 0]

6 Mar 2023

We thank the reviewers for thoughtful suggestions and insights. The manuscript has benefited from these insightful suggestions. We look forward to working with the reviewers to move this manuscript closer to publication in the PLOS ONE. 

The manuscript has been rechecked and the necessary changes have been made in accordance with the reviewers’ suggestions. The responses to all comments have been prepared and attached herewith. (Response to Reviewers) 

We hope that the responses have addressed all the reviewers’ concerns satisfactorily.

---

## [Decision Letter · Decision Letter 1]

20 Mar 2023

Anti-tumor necrosis factor-alpha monoclonal antibody suppresses colorectal cancer growth in an orthotopic transplant mouse model

PONE-D-23-01681R1

Dear Dr. Hayashi,

We’re pleased to inform you that your manuscript has been judged scientifically suitable for publication and will be formally accepted for publication once it meets all outstanding technical requirements.

Kind regards,

Kenji Fujiwara, PhD, MD

Academic Editor

PLOS ONE

Additional Editor Comments (optional):

Dear Dr. Hayashi.

Thank you for submitting the revised manuscript. I think the authors responded well to the opinions of two reviewers. Unfortunately, one previous reviewer did not accept the review for this revision, but instead of the reviewer, I judge the authors' responses are appropriate. I think the manuscript is eligible to be accepted.

Best regards,

Kenji Fujiwara

Reviewers' comments:

Reviewer's Responses to Questions

**Comments to the Author**

1. If the authors have adequately addressed your comments raised in a previous round of review and you feel that this manuscript is now acceptable for publication, you may indicate that here to bypass the “Comments to the Author” section, enter your conflict of interest statement in the “Confidential to Editor” section, and submit your "Accept" recommendation.

Reviewer #2: All comments have been addressed

2. Is the manuscript technically sound, and do the data support the conclusions?

Reviewer #2: Yes

3. Has the statistical analysis been performed appropriately and rigorously? 

Reviewer #2: I Don't Know

4. Have the authors made all data underlying the findings in their manuscript fully available?

Reviewer #2: Yes

5. Is the manuscript presented in an intelligible fashion and written in standard English?

Reviewer #2: (No Response)

6. Review Comments to the Author

Reviewer #2: (No Response)

7. PLOS authors have the option to publish the peer review history of their article (what does this mean?). If published, this will include your full peer review and any attached files.

Reviewer #2: No

---

## [Editor Report · Acceptance letter]

21 Mar 2023

PONE-D-23-01681R1 

Anti-tumor necrosis factor-alpha monoclonal antibody suppresses colorectal cancer growth in an orthotopic transplant mouse model 

Dear Dr. Hayashi:

I'm pleased to inform you that your manuscript has been deemed suitable for publication in PLOS ONE. Congratulations! Your manuscript is now with our production department. 

Kind regards, 

on behalf of

Dr. Kenji Fujiwara 

Academic Editor

PLOS ONE